# Creatures of the state? Metropolitan counties compensated for state inaction in initial U.S. response to COVID-19 pandemic

**Christof Brandtner**[1,2,3,4]*, **Luís M. A. Bettencourt**[4,5], **Marc G. Berman**[6], **Andrew J. Stier**[6]

**1** EM Lyon Business School, Écully, France, **2** Department of Sociology, University of Chicago, Chicago, Illinois, United States of America, **3** Center on Philanthropy and Civil Society, Stanford University, Stanford, California, United States of America, **4** Mansueto Institute for Urban Innovation, University of Chicago, Chicago, Illinois, United States of America, **5** Department of Ecology and Evolution, University of Chicago, Chicago, Illinois, United States of America, **6** Department of Psychology, University of Chicago, Chicago, Illinois, United States of America

* brandtner@uchicago.edu

**Data Availability Statement:** All relevant data can be found on the GitHub repository of the Mansueto Institute for Urban Innovation at https://github.com/mansueto-institute/plosone_covid_counties.

## Abstract

Societal responses to crises require coordination at multiple levels of organization. Exploring early efforts to contain COVID-19 in the U.S., we argue that local governments can act to ensure systemic resilience and recovery when higher-level governments fail to do so. Event history analyses show that large, more urban areas experience COVID-19 more intensely due to high population density and denser socioeconomic networks. But metropolitan counties were also among the first to adopt shelter-in-place orders. Analyzing the statistical predictors of when counties moved before their states, we find that the hierarchy of counties by size and economic integration matters for the timing of orders, where both factors predict earlier shelter-in-place orders. In line with sociological theories of urban governance, we also find evidence of an important governance dimension to the timing of orders. Liberal counties in conservative states were more than twice as likely to adopt a policy and implement one earlier in the pandemic, suggesting that tensions about how to resolve collective governance problems are important in the socio-temporal dynamic of responses to COVID-19. We explain this behavior as a substitution effect in which more urban local governments, driven by risk and necessity, step up into the action vacuum left by higher levels of government and become national policy leaders and innovators.

## Introduction

Societal responses to novel crises require coordination of decisions and action at multiple levels of social organization, usually driven from the top-down. The lack of fast or adequate response at the highest levels, however, creates singular challenges that expose where capacity exists to ensure systemic resilience and recovery in human societies. In this paper, we investigate an instance of local action that explores subsidiary spaces for the emergence of social and political capacity by regional actors, nested within higher-level organizations.

**Funding:** Christof Brandtner acknowledges support by National Science Foundation Soc-DDRI grant #1801677. Marc G. Berman acknowledges support by National Science Foundation S&CC grant #1952050.

**Competing interests:** The authors have declared that no competing interests exist.

We examine the case of sub-national governments—including counties, cities, and states—taking the lead to mitigate the effects of the COVID-19 epidemic relative to higher levels of government. Early research has mainly focused on initiatives led by national governments and by U.S. states [1, 2]; some work even actively omits county-level policies in order to isolate the effects of state policies [3, 4]. Analyzing the diffusion of such local-level policies, our study reveals important structural dynamics through which urban authorities compensate for state policy inaction and lead in the design and implementation of solutions, contradicting the common conception that cities are but "creatures of the state" [5, 6].

To understand how features of local governments and their relationship to higher-level governments can affect compensatory local action, we examined the process by which counties came to adopt COVID-19 containment measures prior to their states. We asked: *Why do some counties pass stand-alone policies ordering people to shelter in place, and why do some pass them sooner than others*? Examining this question, we conducted an event history study of all U.S. counties, estimating the timing of 163 county policies. Drawing on literature in urban and organizational sociology and urban science, we argue that counties' structural position—their rank in the urban hierarchy and their political relationship to their state—explains spatiotemporal disparities in local shelter-in-place orders.

Using daily data on counties in the United States, we study the adoption of local, county-level shelter-in-place orders *prior* to state and national policies as an instance of local exercise of autonomy. Event history analyses show that cities adopted shelter-in-place orders to contain the transmission of COVID-19 the earliest, even as larger metropolitan areas may have experienced the pandemic most intensely early in the process [7]. A substantial gap, on average of 3.5 to 6.6 days, meant that counties with proactive governments were quicker to take measures to contain the spread of COVID-19.

We find that this gap in issuing shelter-in-place orders was not randomly distributed in the US, but followed geographic patterns shown in Fig 1. Both local jurisdiction's position in the social structure and inter-regional contagion influence their relationship to the governments of higher-level jurisdictions in which they are located. Liberal counties in conservative states were twice as likely to adopt a policy early on as counties whose politics aligned with the state. This finding suggests that contrasting beliefs about how to resolve governance problems between cities and their institutional and geographic context explains local exercises of discretion.

These findings are of potential significance for both urban science and public policy. Since the efficacy of containment measures is critically dependent on timing, even small differences of adoption speed—of distancing measures, school closings, and other policies intended to contain the spread of a disease—can accumulate to significant disparities in the number of cases and deaths [2, 8, 9]. As the now-famous comparison between reticent St. Louis and outgoing Philadelphia during the 1918 influenza pandemic illustrates, collective decision-making and action are critical tools for controlling pandemics in the absence of widely available treatments or vaccines [10]. In the conclusion, we discuss our study's implications for understanding the creation and diffusion of local policies and their relationship to higher levels of governance, including a discussion of the mechanisms of the origins of collective social innovation in multi-level political systems.

## Urban autonomy in policy adoption

Local jurisdictions sometimes have discretion to solve governance problems that are in the domain of higher levels of government [11]. For instance, city, regional, and state governments have often taken responsibility for addressing challenges of climate change while national

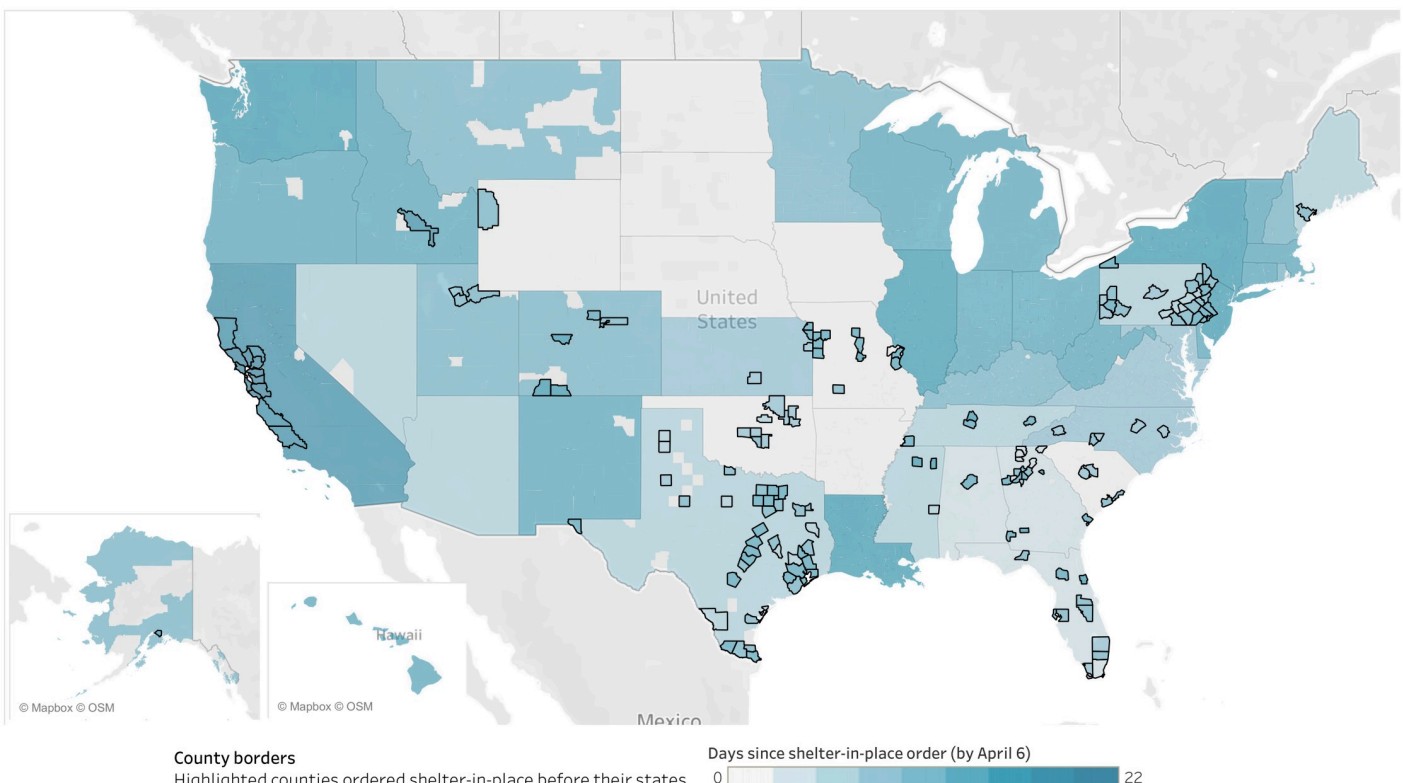

**Fig 1. Map of the USA displaying county-level shelter in place orders.** Showing in darker colors the states and counties that issued shelter-in-place orders in response to the COVID-19 outbreak sooner. Many counties implemented shelter-in-place orders before their states, shown with outlined borders. Map illustrates author's data using base map from Tableau, Mapbox, and OpenStreetMap made available under the Open Database License (https://www.openstreetmap.org/copyright).

governments have embraced such actions more slowly and reluctantly [6, 12, 13]. Cities also tend to be first movers on many other challenges of social collective action, including school reform, civil rights expansion, and policies intended to reduce social inequality [14, 15] as well as in the provision of novel infrastructure and services [16]. From the perspective of urban sociology, such geographic disparities among cities are due to the fact that social problems are experienced dissimilarly in cities of different sizes and that coalitions of interested groups differ from place to place. However, organizations with relatively similar levels of exposure to problems can still differ in their likelihood to conceive and take up solutions and in their basic beliefs about what these solutions should be [17, 18]. Institutional conditions, such as ideological differences and the presence of particular organizational structures, can thus lead to spatial and temporal heterogeneity in the invention and diffusion of practices and policies [19, 20].

Among others, beliefs about the appropriate scope of state intervention can deviate from the dominant beliefs on other levels of government. These beliefs, along with general notions about the "res publica" of local governance, constitute so-called *governance configurations* that set the stage for likely local action [21, 22]. If these basic notions—such as whether the curtailment of individual freedoms is justified in order to protect the collective from harm, or whether the state is responsible for encouraging collective action in light of a public health crisis—are at odds, "governance gaps" [23] can emerge among different actors, including supra-regional governments. Local governments, often by necessity, tend to bridge these gaps in large-scale planning and policy-making in order to maintain a capacity to act [21].

We examine factors that enable local governments in the U.S. to exercise authority that may precede, or diverge from, the actions of higher-levels government. As Peterson [24] famously argued, cities are often limited by higher-level governments—in the U.S., some states limit local autonomy through preemption, citing that cities are but "creatures of the state." Scrutinizing this claim, we suggest that spatiotemporal differences in autonomous policy adoption are due in part to local jurisdictions' social and political environments. In particular, the relationship to higher-level governments as well as local governance configurations are likely better predictors of local action than internal features such as secular political attitudes, budgets, or personalities [21, 22].

Following the research tradition of viewing cities as embedded in a wider social and political structure, we suggest that local governments can replace stifled and inactive higher-level governments [11, 25]. In time, they can champion, create, and perfect innovative policy solutions that may eventually be adopted by higher levels of government—suggesting a basic definition of "urban policy innovativeness" [26]. This argument draws on a structural understanding in which local governance is embedded in higher-level polities, the latter of which have been shown to differ in their degree and speed of innovativeness over time [27]. It is also consistent with Frug and Barron's [28] analysis of how states can "stifle urban innovation" through regulatory and fiscal bounds, driving cities to engage with policy fields in which states show relative inaction and are thus less likely to limit the leeway of mayors and county officials.

## COVID-19 containment and urbanicity

To investigate how the structural position of local jurisdictions affects their propensity to act independently, we examined the conditions under which local governments adopted public policies intended to curb the transmission of COVID-19 during the initial outbreak in the United States. Previous research has highlighted political dynamics at the state level. Adolph and colleagues show that the timing of state-level social distancing policies—including gathering restrictions, mandated school closures, restaurant restrictions, non-essential business closures, and stay-at-home orders—was highly variable and primarily followed partisan lines [2]. States with a Republican governor and a high share of Trump voters showed an additional delay in mandating social distancing of over 2.5 days. These findings are also consistent with geographic differences in the political preferences of individuals. An analysis of GPS data and a survey conducted by Alcott et al. [29 p15] suggest that "individuals' beliefs related to COVID-19 are strongly associated with their social distancing behaviors." Furthermore, mask use in the United States is contingent on partisanship [30]. We extend and qualify these analyses of states and individuals by examining what explains the determinants of *county-level* COVID-19 social distancing measures that preceded state policies.

A direct application of the "political" line of inquiry from states to counties would suggest that more Democratic counties are more likely to pass legislation prior to their respective state. Voting behavior in Presidential elections has indeed been shown to be predictive of the adoption of such policies as legislation opposing fracking [31] and energy-efficient construction [32]. Our theoretical framework, however, suggests that even accounting for a county's political orientation, a suite of additional structural features related to the autonomy of urban governance matter. These structural features are related to cities' role in society and state politics rather than dynamics internal to a city's or county's administration and government.

First, large metropolitan counties are likely marked by greater political and financial autonomy and greater capacity to act than smaller, less urban counties. This is because more, and informationally more sophisticated, organizations tend to be highly concentrated in larger

cities. As a result, such places have more extensive civic infrastructures that in turn enable a greater capacity to act on the behalf of the population [33–35]. For instance, Brandtner and Suárez [6] show that municipalities with a greater number of nonprofit organizations in their organizational ecosystem are more likely to adopt a host of policies and practices related to sustainability. Furthermore, research has shown steep positive associations between population size and other features of urban areas. Such size dependence means that larger cities have, for instance, greater per capita patent activity [33], greater per capita economic activity [36, 37], and more extensive economies of scale of local infrastructure [16] that leads to greater sustainability and faster diffusion of information [34, 38].

Scant research has applied this logic to the adoption of potentially contentious government policies. The gains in information diffusion associated with infrastructural economies of scale brings the potential for faster creation and adoption of time-sensitive policies in larger cities. In line with such a prediction, organizational theory also suggests that larger cities tend to have more extensive and complex administrative apparatuses with heightened bureaucratic capacity. To wit, formal organizations tend to have professional positions and structures that can make them more receptive to ideas and practices and can circulate these novel ideas to their wider environment, such as HR professionals making firms attuned to anti-discrimination policies [39] or more bureaucratically advanced government units being more receptive to new methods of formal organization [40, 41]. In combination, the heightened organizational capacity and pace of coordination among dissimilar governance actors suggest that larger counties, and those that form the whole or a part of a metropolitan statistical area, are more likely to adopt a stand-alone COVID-19 policy, all else being equal.

Second, following the above discussion of how cities may compensate for state inaction in a crisis, urban counties in states displaying little to no willingness to pass containment legislation may become most proactive. This prediction mirrors the logic that particular organizational forms substitute for the failure of other spheres of influence. In the case of market failures, firms' internal structures compensate for the weaknesses of the market in difficult transactions [42, 43]. In the case of government failure, nonprofit organizations compensate for the inability of the state to serve fringe groups among its constituents [44]. Similarly, when nation states and other higher-level governments fail to fill a local need, local jurisdictions can step up and compensate for such "governance failures" [45].

How governance failures are resolved is shaped by the cultural and political alignment between local and state governing bodies. Like state and national politics, municipal politics can vary ideologically, which is consistent with differences in both the population's and politicians' beliefs about how challenges are best tackled [46]. Given the reluctance of some states to adopt new practices and policies, the governance gap between cities and states creates a vacuum for local leaders to fill [23, 26, 47]. In the particular political economy of COVID-19 in the United States, this would suggest that progressive-leaning counties that are embedded in more conservative states tended to adopt stand-alone policies sooner.

Third, like other state organizations, counties and municipalities rarely make decisions in a vacuum, but instead in relation to peer jurisdictions [39, 48]. For instance, Martin [14] showed that cities adapt their living wage legislation to redistributive municipal policies among peer cities and Steil and Vasi [15] showed patterns of policy contagion among sanctuary cities. As Tolbert and Zucker [49] demonstrated, policies that are initially adopted for functional reasons—such as civil service reform aimed at curbing corruption in city administrations—can later become institutionalized and considered a desirable default that is adopted regardless of the policy's function. Legitimacy effects suggest that horizontal inter-organizational diffusion complements other established explanations for the uptake of urban innovation such as city size [50]. Relatedly, a series of studies on the diffusion of state lotteries, economic development

plans, and strategic planning among municipal and state organizations show that U.S. public administrations emulate peer organizations [51, 52]. Following this line of work, we expect that local shelter-in-place orders create local spill-overs and even establish a default policy in states with many different local policies.

These factors are argued to have a discrete influence on local decisions even controlling for other economic and demographic county features. Among others, we expect that county leaders are more likely to adopt stand-alone policies if they are at a higher risk of COVID-19 outbreaks and associated deaths. This is again true for larger, denser cities [7, 53] and if the population is particularly vulnerable.

## Materials and methods

We conducted event history analyses of the time to the first adoption of a county-level shelter-in-place order in all U.S. counties since the first order was placed on March 16 using Cox models [2, 54]. The outcome variables are based on a publicly available New York Times investigation into shelter-in-place orders and coding by Painter and Qiu [4] and the authors. We combined these data with county-level data from the American Community Survey [55], David Leip's Atlas of Presidential Elections [56], a state policy innovativeness indicator and other state-level policy factors from the Correlates of State Policy project, and health indicators from the Robert Wood Johnson County Health Rankings [27, 57]. Table 1 reports on the sources and construction of all variables in greater detail.

### Sample

We examined the timing of shelter-in-place orders at the county-level since the onset of the COVID-19 pandemic in the United States on January 21. Event history models estimate how different county and state features affect a county's propensity of adopting a policy on any given day. The hazard set for these models is the entirety of U.S. counties between the first record of a case of COVID-19 in the United States on January 21 and April 5, when no new counties issued new shelter-in-place orders. The first failure (i.e., adoption of a shelter-in-place order) in our data was March 16, when seven San Francisco Bay Area counties became the first to issue a shelter-in-place order effective March 17. Because we were interested in the issue of orders rather than their effect, we determined a failure based on the date on which an order was issued, and not when it went into effect; typically, orders went into effect on the same or the following day.

### Dependent variables

The shelter-in-place policies of counties and states, tracked on the county-level, come from Painter and Qiu [4], who coded the data from the New York Times [58] coverage of sub-national shelter-in-place policies. We then added to these datasets through systematic manual searches. We describe our search procedures in the appendix, together with a list of all 163 counties that passed shelter-in-place orders prior to their respective state. The primary outcome reported is whether a county had a shelter in place policy, but no state-level order, at any given day. We also estimated the timing of adoption of *any* shelter-in-place policy, regardless of whether it was issued by the county or the state, to provide a baseline estimate of what models would look like that disregard local autonomy.

In secondary analyses, we also investigate the association between the issuing of county- and state-level policies and subsequent social distancing and COVID-19 growth rates. For social distancing, we drew on the Unacast Social Distancing Scoreboard [59]. These data included daily measures of the reduction of: a) average distance traveled and b) visits to

**Table 1. Descriptive statistics of key variables for county-days.**

| Variable | Source & variable construction | N | Mean | S.D. | Min | Max |
|---|---|---|---|---|---|---|
| County order (dummy) | New York Times, Painter & Qiu [4], author's coding | 174,048 | 0.01 | 0.09 | 0 | 1 |
| Any order (dummy) | New York Times, Painter & Qiu [4], author's coding | 174,048 | 0.10 | 0.29 | 0 | 1 |
| County order (date) | New York Times, Painter & Qiu [4], author's coding | 11,618 | 24 Mar | 3.66 | 16 Mar | 2 Apr |
| Any order (date) | New York Times, Painter & Qiu [4], author's coding | 89,313 | 28 Mar | 4.85 | 16 Mar | 7 Apr |
| Population | Logged population as per 2010 Census | 174,048 | 10.71 | 1.32 | 6.60 | 16.13 |
| Population density | Population / county surface area from Census Gazetteer files | 174,048 | 4.27 | 1.52 | -3.23 | 11.18 |
| County orders in state | State share of counties with a county order (except focal county) | 174,048 | 0.01 | 0.04 | 0.00 | 0.39 |
| County share Democrats in 2016 Presidential Election | County's voter share Clinton in 2016 presidential election (Leip's Atlas [56]) | 174,048 | 0.34 | 0.15 | 0.07 | 0.88 |
| County vs. state share Democrats in 2016 Presidential election | Difference between state's voter share Clinton and county's vote share Clinton in 2016 presidential election (Leip's Atlas [56]) | 174,048 | -0.06 | 0.14 | -0.42 | 0.53 |
| State preemption | Number of policies in which states preempt municipal governments from action (National League of Cities) | 174,048 | 3.21 | 1.38 | 0 | 6 |
| State policy innovativeness | Indicator of early policy adopting states from Boehmke and Skinner [27] | 174,048 | 0.09 | 0.17 | 0.00 | 0.86 |
| Median income | American Community Survey (5-year estimates) | 174,048 | 10.84 | 0.26 | 9.91 | 11.82 |
| Income inequality | American Community Survey (5-year estimates) | 174,048 | 0.45 | 0.04 | 0.35 | 0.66 |
| Families below poverty line | American Community Survey (5-year estimates) | 174,048 | 11.32 | 5.41 | 0.40 | 52.10 |
| Percent high school graduates | American Community Survey (5-year estimates) | 174,048 | 33.84 | 7.42 | 8.10 | 55.60 |
| Percent race white | American Community Survey (5-year estimates) | 174,048 | 81.72 | 16.80 | 9.10 | 99.70 |
| Median age | American Community Survey (5-year estimates) | 174,048 | 40.35 | 5.04 | 21.50 | 66.00 |
| Care physicians per 1,000 capita | Robert Wood Johnson Foundation's County Health Rankings | 174,048 | 0.55 | 0.33 | 0.00 | 4.53 |
| Percent uninsured | Robert Wood Johnson Foundation's County Health Rankings | 174,048 | 0.12 | 0.05 | 0.02 | 0.33 |
| First COVID-19 case | First occurrence of positive test in county (NYT) | 174,048 | 21 Mar | 8.22 | 21 Jan | 4 Apr |
| Cum. cases of COVID-19 | Logged number of cases based in county (NYT) | 174,048 | 0.58 | 1.47 | 0.00 | 13.10 |
| COVID-19 growth rate | Three-day rolling average of COVID case growth | 174,048 | 0.07 | 0.18 | 0 | 4 |
| COVID-19 testing | Logged per capita number of tests administered in state based on data from The Atlantic's COVID Tracking Project | 174,048 | -7.07 | 4.80 | -16.13 | 6.30 |
| Distancing | Principal component factor of percent change in average distance traveled and percent change in essential locations visited from Unacast Social Distancing Scoreboard [59] | 76,301 | -0.03 | 0.98 | -5.18 | 3.52 |

Including source and variable construction. All coefficients standardized in regressions.

non-essential places. To combine these two highly correlated variables (r = .76), we created a factor variable of visits and travel distance. The Cronbach's alpha for this factor is .86, the Eigenvalue of the first factor is 1.76 and Eigenvalue of the second factor is .23; both distancing measures loaded on the first factor with an equal weighting of .94. To calculate the COVID-19 growth rates, we created a three-day moving average of daily growth in the number of cumulative cases by county. We also calculated the day of maximum growth during the study period based on this measure.

## Independent variables

A large variety of datasets inform our model of the covariates of local shelter-in-place policies: (a) *Demographic variables* came from the American Community Survey (2018 5-year averages) and the 2010 Census. These measures include population, which we logged because of skewness, and population density based on the population by square mile drawn from the Census

Bureau's 2017 gazetteer files. (b) To measure *contagion*, we include the share of counties within the same state that had previously adopted a shelter-in-place policy. The measure is based on summary statistics of our dependent variable excluding the focal county. We opted for a relatively coarse contagion measure because the short time span of the initial policy adoption makes it difficult to hypothesize and test a sequenced process of county-to-county diffusion. (c) *Political variables* were based on 2016 Presidential election data from David Leip's [56] Election Atlas. The difference in election results is calculated by subtracting the population-weighted average from the county's election result; a positive value means that voters in the focal county were more likely to vote for Hilary Clinton than the state as a whole. State pre-emption was measured as an index of policy areas in which States preempted local governments according to the National League of Cities. State policy innovativeness was measured via an index from Boehmke and Skinner [27], which extends work by Walker [26]. (d) The *socio-economic status* of the county, including median household income, Gini of income distribution, poverty level, education, race, and median age, was also based on data from the American Community Survey. (e) Two measures related to *health* came from the Robert Wood Johnson 2018 County Health Rankings: the number of care physicians per capita and the share of people without insurance. (f) We also controlled for features specific to *COVID-19*. The date of the first COVID-19 case was recorded in the county to control for differences in the onset of the pandemic. A series of measures indicating the intensity by which COVID-19 was experienced in the county, including the number of positive COVID-19 cases reported in the county, the 3-day average of the COVID-19 case growth rate, and the number of tests per capita administered in the state account for the state's capacity to manage the pandemic.

Our classification of counties by *urbanicity* followed the simplified National Center for Health Statistics (NCHS) urban-rural classification scheme [60]. Large central metros are counties in Metropolitan Statistical Areas (MSA) with 1 million or more population and contain the largest principal city of the MSA, have their entire population contained in the largest principal city of the MSA, or contain at least 250,000 inhabitants of any principal city of the MSA. Large fringe metros are counties in MSAs with 1 million or more population that did not qualify as large central metro counties. Medium and small counties are counties in MSAs of populations with less than 1 million inhabitants. Micropolitan and non-core counties are not located in an MSA.

## Methods

We used Cox models to estimate a county's hazard of adopting a policy prior to the state. The Cox model is suitable for modeling the timing of adoption and, as a semi-parametric model, makes no assumption about the shape of the distribution of hazard rates. The equation $h(t) = h_0(t) \exp(BX)$ specifies the diffusion process, where $h(t)$ is each county's hazard of issuing an order in time $t$, $h_0(t)$ is the baseline hazard at time $t$, $X$ is a set of $k$ county-level covariates, and $B$ is a vector of regression coefficients $\beta_k$ for these covariates. All coefficients are standardized for ease of comparing effect sizes. The full model presented in Table 2 and Fig 2 is expressed by the following equation:

$$h(t) = h_0(t)\exp(\beta_1 X_{demographic} + \beta_2 X_{political} + \beta_3 X_{socio-economic} + \beta_4 X_{health} + \beta_5 X_{COVID-19}) \quad (1)$$

Coefficient plots illustrate how a standard deviation change in each covariate is estimated to affect the hazard of issuing an order at time $t$ [61, 62]. The hazard functions illustrate $h(t)$ over time. In additional models, we split the sample by the dominant politics of the county and estimated $h(t) = h_0(t)\exp(XB_{Democratic})$ separately from $h(t) = h_0(t)\exp(XB_{Republican})$. We also ran

**Table 2. Cox model regression coefficients of variables predicting adoption of county order prior to state order to shelter-in-place during early days of COVID-19 in the United States.**

| | (1) | (2) |
|---|---|---|
| | County order | Any order |
| Population (logged) | .842** | .024 |
| | (.309) | (.202) |
| Population density | .510* | .218* |
| | (.230) | (.087) |
| County orders in state | .267*** | .026 |
| | (.046) | (.034) |
| County share Democrats 2016 | -.182 | .464** |
| | (.256) | (.177) |
| County vs. state share Democrats 2016 | .657** | -.295+ |
| | (.202) | (.162) |
| State preemption | .010 | .069 |
| | (.138) | (.081) |
| State policy innovativeness | -.399 | .045 |
| | (.324) | (.130) |
| Median income | .117 | .174+ |
| | (.183) | (.094) |
| Income inequality | .182+ | .026 |
| | (.100) | (.052) |
| Families below poverty level | -.845*** | .190* |
| | (.180) | (.081) |
| Percent high school educated | -.230 | .012 |
| | (.223) | (.065) |
| Percent race white | .144 | .140 |
| | (.171) | (.089) |
| Median age | -.087 | .060 |
| | (.107) | (.040) |
| Care physicians per capita | .091 | -.017 |
| | (.084) | (.028) |
| Percent uninsured | .779*** | .081 |
| | (.160) | (.064) |
| Date of first COVID-19 case | -.141 | -.525*** |
| | (.110) | (.088) |
| *Time variant* | | |
| Cum. cases of COVID-19 | -.906*** | -.793*** |
| | (.116) | (.078) |
| COVID-19 growth rate | .057 | .056** |
| | (.051) | (.020) |
| Cum. tests per capita in state | -1.881*** | -1.490*** |
| | (.497) | (.452) |
| Observations (county-days) | 174,048 | 174,048 |
| *AIC* | 2689.09 | 36248.30 |
| *df* | 19 | 19 |

All variables X-standardized by z-scoring variable; standard errors in parentheses;

+ p < .1,

* p < .05,

** p < .01,

*** p < .001.

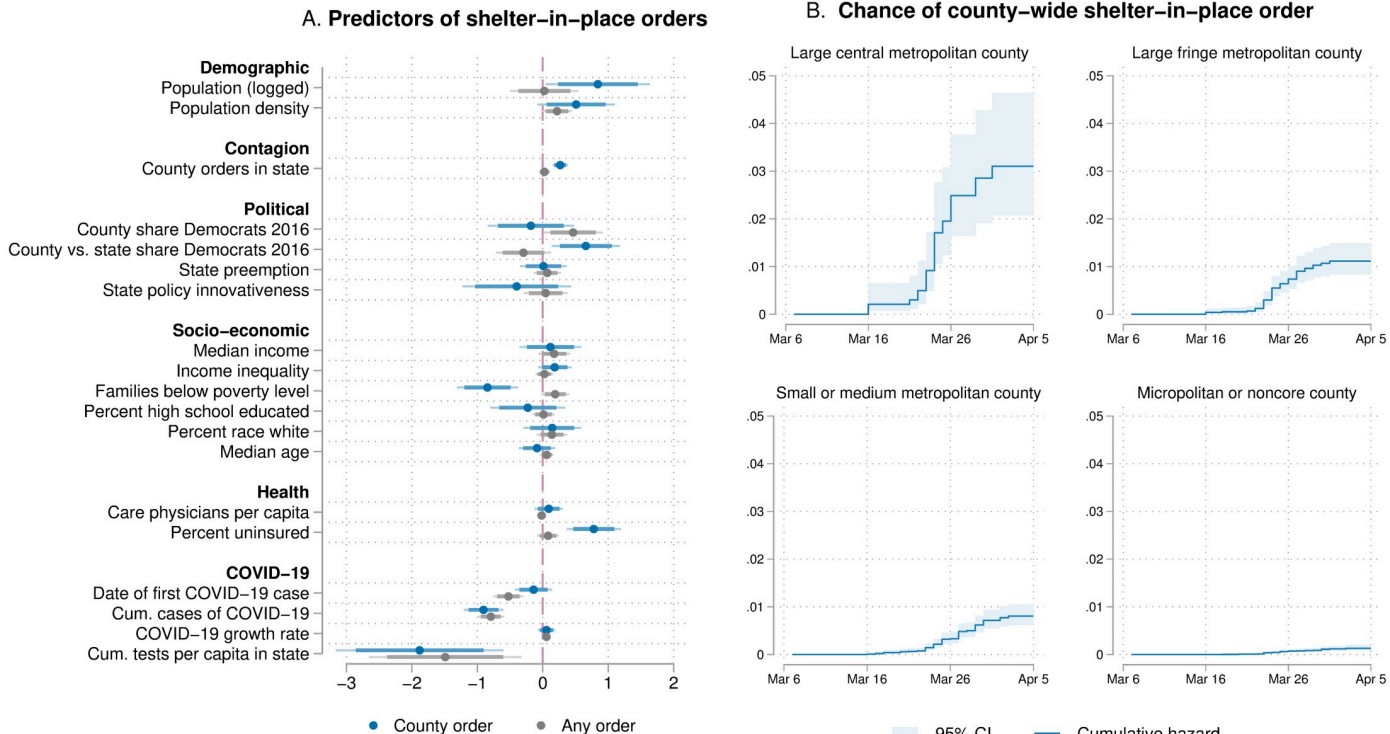

**Fig 2. Predictors of hazard rate of adopting a shelter-in-place order.** Panel A shows the local predictors of shelter-in-place policy responses (in blue), examining the association between demographic, political, socio-economic, and health-related features of counties $X$ and the hazard of passing a shelter-in-place policy $h(t)$ through the equation h(t) = $h_0(t)exp(BX)$. We compare the predictors of county-level shelter in place orders to the predictors of any shelter-in-place order, including state orders, as a control model (in gray). Panel B displays Nelson-Aalen cumulative hazard function by urbanicity, indicating a greater hazard rate among larger, more central metropolitan counties.

models including time-varying measures of the cumulative COVID-19 caseload, the COVID-19 test capacity, and how the COVID-19 growth rate interacted with time. We interacted a vector of coefficients Z with time in order to estimate their time-variant effect ɣ:

$$h(t) = h_0(\text{t})\exp(XB + \gamma Z_{COVID-19}) \tag{2}$$

Since we studied the entirety of U.S. counties, we note that in our study, significance levels indicative of generalizability to other countries are of secondary concern. Significance levels remain indicative of the magnitude of the effect relative to the coefficient's variance.

## Results

We present our findings in four steps. First, we show descriptive statistics demonstrating that several counties adopted policies prior to their states. We then examined which county-level covariates are associated with the timing of county-level orders. We then discuss how these findings relate to the political geography of the United States, noting evidence in support of our hypothesis that metropolitan counties outpace their respective states if they deviate from the state politically. Finally, we show that these findings relate to different trajectories of the first wave of the COVID-19 pandemic.

## Descriptive statistics of county-level shelter-in-place orders

As Fig 1 shows, counties with autonomous shelter-in-place policies closed businesses and ordered the population to stay at home significantly sooner than surrounding counties that did not have such orders. Between March 16 and April 2, a total of 163 counties passed local shelter-in-place orders prior to their states. Although only 5% of all counties adopted a policy, these counties accounted for a total population of 74.8 million people. This means that about 1 in 5 U.S. Americans were ordered to shelter in place by a county rather than a state for at least one day of the study period.

How long is this delay between county and state action? The median date of county orders was March 24, with a range from March 16 to April 2. The median state policy date in counties with policies already in place was April 2; the median date of all state policies was March 30. The resulting time gap between county and state policies was between one and fourteen days (Georgia), with 50% of the orders being four to nine days ahead of their states. On average, local policies were passed some 3.5 days ahead of the average state policy.

## Multivariate analysis of county-level shelter-in-place orders

Fig 2 shows the hazard rates of adopting a local shelter-in-place order. The panel on the right visualizes the Nelson-Aalen cumulative hazard function by urbanicity. The hazard plot indicates that larger and more metropolitan counties had a notably higher hazard rate throughout the critical adoption period. The event history analysis reported in the left-hand panel shows that this pattern is statistically significant. Trend lines and coefficients presented in blue indicate counties' shelter-in-place orders. Counties that are a single standard deviation larger are more than four times as likely to adopt a local shelter-in-place order as a less populous county at any given point in time during the study period (hazard rate = $e^{.84}$ = 2.32, p < .001, 95% CI: 1.27 to 4.26). The control model, depicted in gray, uses the same regressors as the previous model to predict when a county is affected by *any* shelter-in-place order, including a state mandate. This model shows that larger counties with greater population density were more likely to see any form of policy, including state policies, which implies that states with denser urban centers also tended to adopt state-level policy orders (p < .05). The demographic covariates suggest that this association may be due to the economic capacity of local states, as counties with higher poverty rates by one standard deviation were half as likely to adopt a local shelter-in-place policy ($\beta$ = -.85, p < .001).

Our results further indicate that voting for Hilary Clinton over Donald Trump in the 2016 presidential election was not the best predictor for a county's shelter-in-place order. In fact, according to our analysis, political preference was not a statistically significant predictor of a local shelter-in-place policy (p > .05), although counties with Democratic voters were marginally more likely to see any policy at all—likely driven by the state-level orders of Democratic-leaning states (p < .01). These findings reinforce our contention that the direct extension of political arguments about state-level COVID-19 interventions to the county-level is inappropriate because it neglects governance dynamics of local pandemic responses.

Our models show a strong association between local shelter-in-place orders and a county's political context. Counties where the difference between local and state-wide election results was greater by a single standard deviation were 1.93 times more likely to adopt a policy at any given point in time ($\beta$ = .66, p < .01). Fig 3 shows how the difference in election results is distributed in the United States (shade of color) and how local shelter-in-place orders map onto this political geography of the country (bold county borders).

To be sure, not all counties that passed a policy are more Democratic than their state. For example, in the State of Pennsylvania, the governor ordered shelter-in-place in some counties

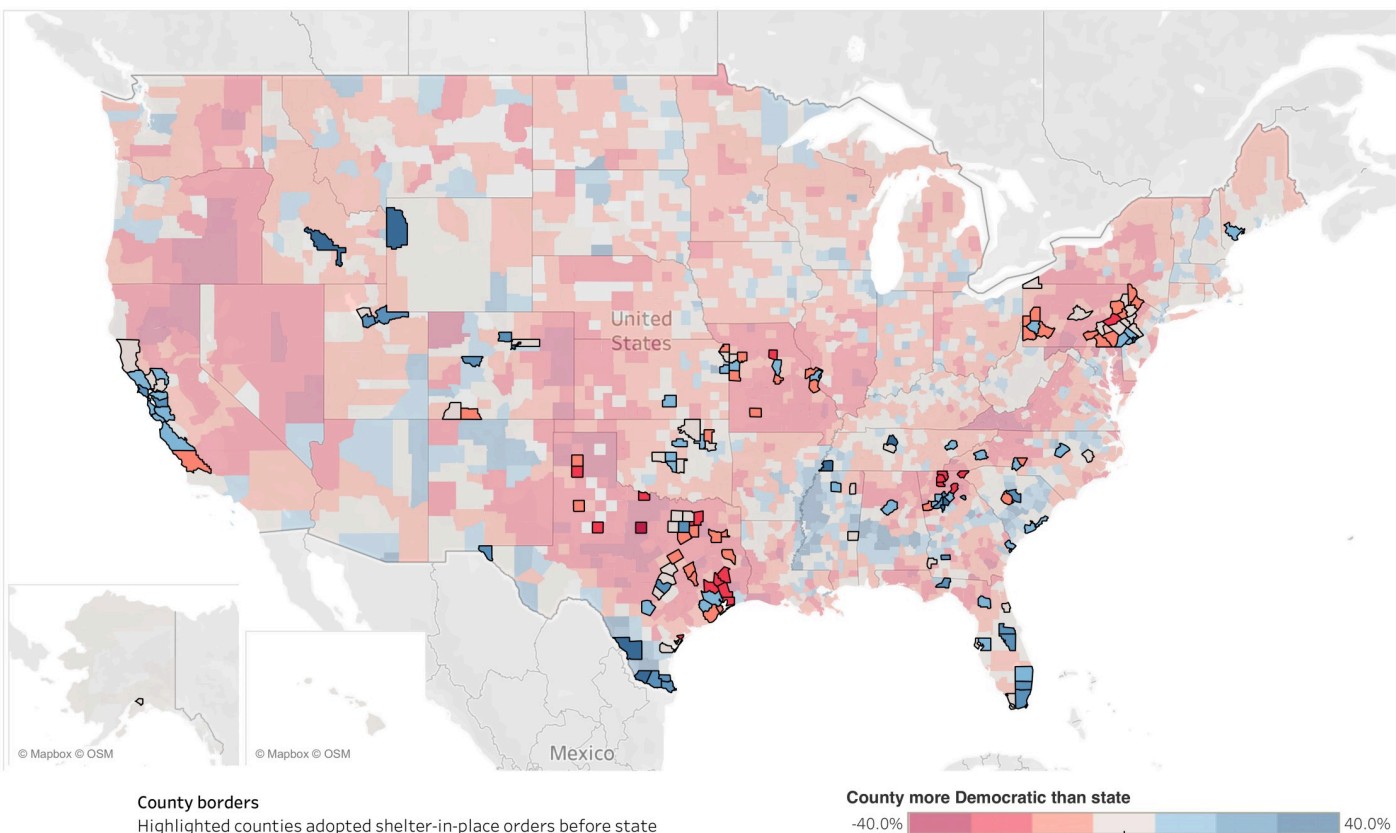

**Fig 3. Distribution of county shelter-in-place orders by democratic vote share.** Map of the contiguous United States showing that a county vote share from the 2016 Presidential election that contrasted with its state average was predictive of county shelter-in-place orders occurring before the state's shelter-in-place orders. Counties that passed shelter-in-place policies prior to their states have highlighted borders. The majority of local orders were passed in counties that were more Democratic than their state or spatially proximal to such a county. Map illustrates author's data using base map from Tableau, Mapbox, and OpenStreetMap made available under the Open Database License (https://www.openstreetmap.org/copyright).

and in Texas, several Republican-leaning counties in proximity to Dallas passed orders earlier than the state. Our findings are robust to the exclusion of these states, as including them attenuates the effect of the vote share difference.

Two control variables shed further light on the dynamics of local adoption. The first is the number of policies adopted by other counties in the same state—a basic measure of contagion among counties—, which we found to be significant ($\beta = .27$, $p < .001$). Second, we found that counties with a greater share of uninsured individuals were almost twice as likely to see a shelter-in-place order, which suggests that policy makers may be responding to high levels of vulnerability in their population ($\beta = .78$, $p < .001$). Naturally, counties that were hit by COVID-19 earlier tended to also see shelter-in-place policies sooner ($\beta = -.79$, $p < .001$), but this association does not hold for county-level orders alone ($p > .05$). The timing of county-level orders was thus *not* a function of denser and more Democratic-leaning counties simply encountering positive cases of COVID-19 before more rural, Republican-leaning areas.

These findings hold true despite the inclusion of a large number of controls about county demographics and economics, state policy regimes, and exposure to COVID-19. As with state-level policies, there are statistically significant associations with the daily change in the number of total cases and the average growth rate indicative of temporal patterns. But most orders were issued when the COVID-19 case prevalence was still small ($\beta = -1.88$, $p < .001$). The

negative coefficient confirms Adoloph et al.'s [2 p12] earlier insight that "while many of the early epicenters were Democratic-leaning states, it does not appear that the more aggressive action of Democratic states is simply a function of caseload."

## Unpacking the compensation effect

A secondary model predicting the likelihood of local shelter-in-place orders, presented in Fig 4, helps unpack the governance dynamics revealed by the statistically significant effect of political differences between counties and states. Separating majority-Democratic from majority-Republican states, we examined whether our finding was driven by liberal counties in conservative places being proactive, or by conservative counties in liberal states hanging back.

The Cox model of having adopted a local shelter-in-place order shows that Democratic-leaning counties are almost three times as likely to adopt a shelter-in-place policy than other Democratic-leaning counties if the degree to which they are more liberal than their surrounding state is one standard deviation greater (β = 1.00, p < .001). Among Republican-leaning counties, there is no significant effect (p > .05). It is also notable that the date of first encountering COVID-19 is a strong predictor of county-level policies among Republican counties (β = -.53, p < .001). The interpretation that the timing of policies is a result of the trajectory of

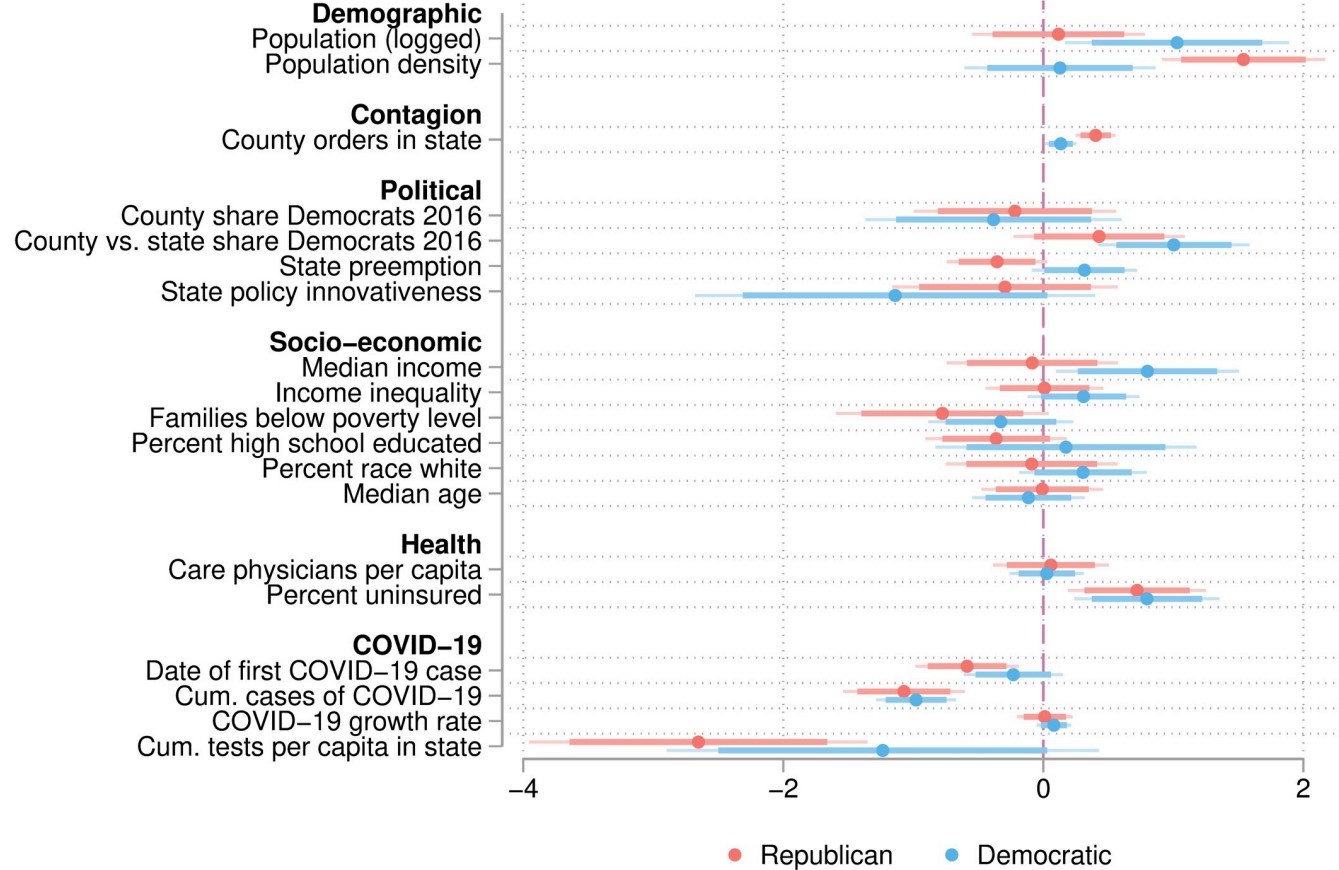

**Fig 4. Regression coefficients of split-population Cox models by politics.** Predicting presence of a county-wide policy in predominantly Republican (N = 2,653) vs. Democratic (N = 489) counties. The regression by political orientation shows that the effect of political difference to the state is driven by Democratic counties in Republican states (rather than Republican counties in Democratic states). In Republican counties, shelter-in-place orders were associated with functional factors: Greater population density, experiencing COVID-19 sooner, and if state capacity to test was lower. The coefficients of percent of uninsured population indicates that population vulnerability increased a county's propensity to issue a policy. Main results of political differences between county and state are stronger if TX and PA are excluded, because in both state leadership was involved in ordering shelter-in-place in select counties.

the pandemic does not hold for their Democratic neighbors, however (p < .05), indicating that political considerations outweighed a forced hand due to the public health urgency.

Instead, these secondary models lend direct support to the argument that greater autonomy was exercised primarily among Democratic counties that faced an inactive or even restrictive governance context on the state-level. The effect of a state's tendency to pre-empt local juris-diction differed depending on the political leaning of the county. Whereas Republican counties were more likely to pass legislation in states with *lower* state preemption (β = -.36, p < .05), Democratic counties were more likely to pass legislation in states with *greater* state preemption (β = .32, p < .05). Finally, we note that the prevalence of more COVID-19 tests in the state had a muffling effect, suggesting that it was particularly counties in states that were less proactive about the pandemic that chose to pass shelter-in-place orders on their own.

These findings also illuminate the question whether liberal and conservative policies respond to the vulnerability of their population differently. The share of people without insur-ance in the county has a strong and significant association with local shelter-in-place orders in both Democratic-leaning (β = .79, p < .001) and Republican-leaning (β = .72, p < .001) counties.

## Estimated effect of local shelter-in-place orders

Our data do not allow us to make strong causal claims about the impact of local government policies on the containment of COVID-19. There is, however, existing evidence that the early adoption of county-level policies made a significant difference for the spread of the disease in U.S. states. Dave et al. [8] find that in early adopting Texan counties, "COVID-19 case growth fell by 19 to 26 percentage points two-and-a-half weeks following adoption of a shelter-in-place order." These findings are robust for controls for the outbreak timing, testing regimes, and containment measures in neighboring counties. Dave et al. [8 p4] claim that "this effect is driven nearly entirely by . . . highly urbanized and densely populated counties" and that "the later statewide shelter-in-place mandate yielded relatively few health benefits" in addition to the quick response of what the authors call early-adopting "urban cowboys."

Our national-level findings are consistent with this proposition, suggesting that the adop-tion of a local-level policy is associated with a subsequent decline in distance traveled and places visited. Overall, a county order increases distancing by about 24% (p < .001) controlling for the logged number of COVID-19 cases and the number of COVID-19 tests administered in the state. The associations shown in Table 3 suggest that the progression of the social response to COVID-19 is different in counties that adopted a policy than in counties that did not.

As the left-hand panel of Fig 5 shows, counties with a stand-alone order in place tended to see greater levels of social distancing following the order and throughout the month of April, compared to counties with state orders and counties with no shelter-in-place policy. The time series in the right-hand panel also reveals that such counties tended to have a greater number of COVID-19 cases reported during this period, which is likely associated with the insight that the COVID-19 attack rate is highest in denser, wealthier urban areas [7]. Although these coun-ties are hit harder at the onset, the case growth reaches the epidemic peak sooner in counties that had a policy. Because it remains possible that the relationship between public policies for social distancing and the number of cases and deaths is spurious, we leave it to future work to establish the causal effect of such correlations. For the purposes of this paper, we simply note that counties with and without shelter-in-place orders have a significantly different trajectory in the early days of the pandemic than did other locales.

**Table 3. Ordinary Least Squares (OLS) coefficients of variables predicting policy effects on distancing with county fixed effects.**

|  | (1) |
|---|---|
| Post-county order | .237*** |
|  | (.028) |
| Logged COVID-19 cases | .238*** |
|  | (.014) |
| Logged COVID-19 tests per capita in state | -.081* |
|  | (.032) |
| Constant | .496*** |
|  | (.045) |
| *County fixed effects* | Yes |
| Observations | 2,341 |
| $R^2$ | .336 |
| *df* | 3 |

County-level fixed effects hold time invariant features of counties constant. Results show that there is a significant post-order effect on distancing after a county adopted a shelter-in-place order. Distancing was also greater in places with a higher number of COVID-19 cases, suggesting that counties that were more affected also saw a greater social response. All variables X-standardized by z-scoring variable; standard errors in parentheses;[+] p < .1,

[*] p < .05;

[***] p < .001.

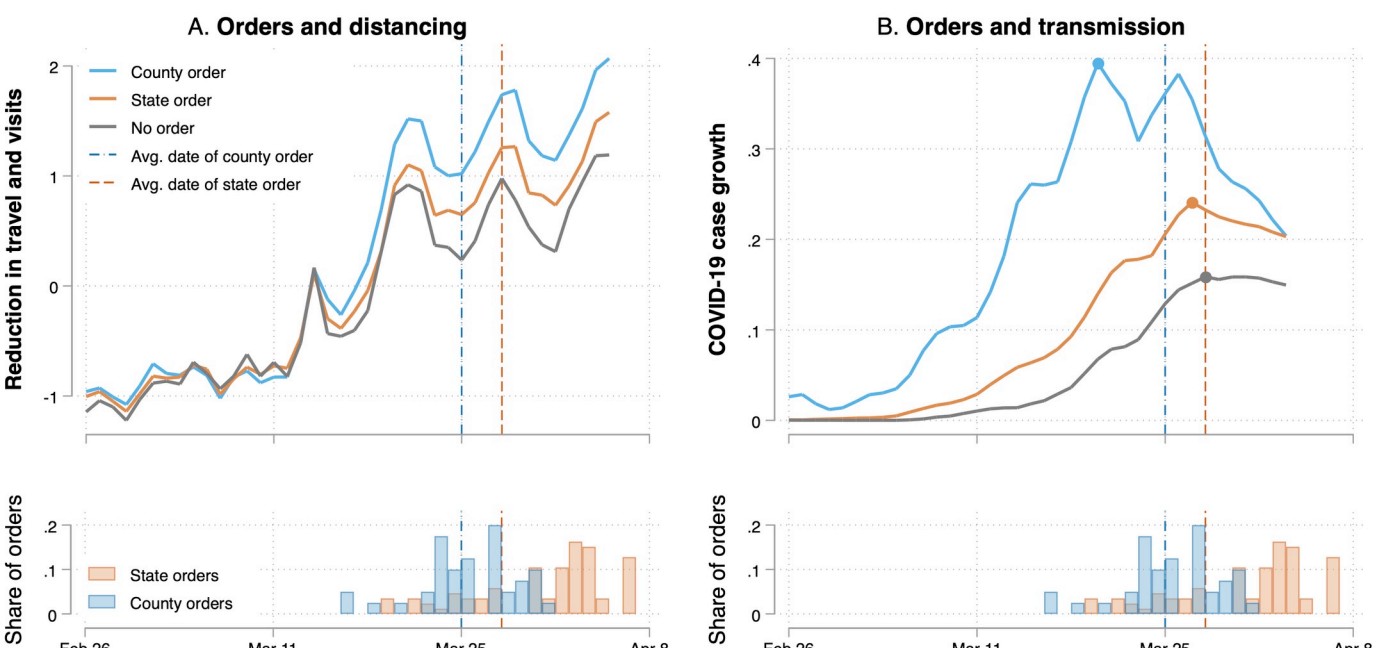

**Fig 5. COVID-19 growth rates and distancing by shelter-in-place order over time.** The Figure shows that counties issuing a policy before their states passed policies sooner, experienced greater distancing measured by reductions in the average travel distance in the county (Panel A), and reached the epidemic peak before other counties (Panel B). This suggests that county-level policies are related to meaningful differences in both social behavior and epidemic transmission, although this effect may be correlational rather than causal. The share and average date of shelter-in-place policies adopted by counties and states respectively is depicted in the bottom panels.

## Discussion

Our analysis of how U.S. counties responded to the first wave of COVID-19 finds that local shelter-in-place orders were primarily driven by larger and presumably more autonomous counties and counties whose population does not share the political preferences of the surrounding state. This result is consistent with the idea of a compensation effect, in which cities step up if silence or refusal to act on the part of a state leadership creates a vacuum of inaction. As our models confirm, multi-level governance dynamics have implications for our understanding of social behaviors in cities.

Although cities' density has disadvantages during a pandemic, there are also political and cultural dimensions that explain, in the words of Jane Jacobs, "the kind of problem a city is" [38]. While the COVID-19 pandemic has exposed natural disadvantages of cities to the spread of an infectious disease, the pandemic also demonstrates that cities have increased capacity to act and are more responsive to urban innovation than other forms of social organization. The association between governance failures and the autonomy of increasingly capacious cities is likely more broadly relevant [21, 23, 45]. Future work could use such governance gaps and divergences in political preferences between counties and states as institutional conditions of the diffusion of other policies such as municipal fiscal incentives in high-tax states, local gun laws in highly restrictive states, or the local embrace of progressive policies to further the rights of LBGTQ people, undocumented immigrants, and other marginalized groups [15, 18, 19, 31].

Our findings speak to the ongoing debate about the autonomy of cities vis-à-vis their regional or state governments [6, 11, 28]. In the United States, states differ in the degree of freedom they allow their cities, often imposing limits on urban autonomy [24]. We find that restrictive states that actively preempt local action in some domains can actually create the opposite effect in the context of public health: cities innovate to compensate for the absence of state action and may set precedents for their action. Other examples of such a compensation effect include liberal urban climate change policies and public health initiatives in conservative or less environmentally aware states and nations. Although we are convinced that the idea of cities compensating for the limitations of states and nation states is generalizable to other countries, the relationship between cities and higher-level governments is not internationally uniform. We would expect compensation effects to be more pronounced in highly polarized countries, or where the political preferences between cities, regions, and nations diverge to a greater degree. We also expect compensation related to market regulation and fiscal policies to be more prevalent in liberal market economies compared to coordinated market economies in which federal states take responsibility for the provision of social services.

To be sure, the present analysis is limited to the early adoption of shelter-in-place policies, based on the assumption that immediate action had an outsized impact on the spatiotemporal trajectory of the disease progression. It is likely that these policies correlate highly with other measures to contain the spread of COVID-19 taken later on. Our data do not account for differences in the policy regime after April 5, however. Future work could explicitly examine the long-term consequences of early shelter-in-place policies, for instance by creating interdependencies between municipal and state policies (such as increased collaboration between the mayors of large cities and state governors) and by creating path dependencies from past to future responses to COVID-19, such as mask mandates, school reopenings, or stay-at-home policies in later waves of the pandemic [5, 63].

Given the plausibility of long-term consequences of the pandemic on many different socioeconomic dimensions such as race relations, political polarization, and economic development, it is also likely that the autonomous actions of cities can signal, or even fuel, conflict between different political jurisdictions, including the federal level. These tensions may be

positive in the sense of creating laboratories for future policies at the national level, but such experiments may also force disagreements with higher levels. Future work should examine the governance dynamics of the pandemic over longer periods of time and with respect to a broader set of outcomes to test whether the structural position of the county in urban hierarchies has long-term consequences for action with respect to COVID-19 and beyond. Besides government action, there may be so-called "institutional legacies" for the founding of public health cooperatives and economic recovery from pandemic-induced recessions [64, 65].

Despite concerns about the potential tensions among local, regional, and national jurisdictions, this study has revealed a staggering innovative capacity of local actors in responding to COVID-19. Every fifth person in the U.S. was ordered to stay at home by a county before a state took action, and other local governments as well as states eventually emulated proactive counties. The urban status and structural context of counties explains the spatiotemporal pattern of these influential COVID-19 policies. This conclusion adds to mounting evidence challenging the notion of cities as "creatures of the state" and shows that governance configurations matter for how social problems are resolved across multiple levels of political organization.

## Supporting information

**S1 File.**
(DOCX)

## Acknowledgments

We thank Molly King and two anonymous reviewers for helpful comments on earlier drafts of this manuscript. We also thank Patrick Bergemann, Elisabeth Clemens, John W. Hanson, Krystal Laryea, Nicole Marwell, Scott Ortman, Woody Powell, Amanda Sharkey, Nick Sherefkin, Satej Soman, and Austin Wright for their advice. We are further indebted to Olivia Paraschos and Ana Gonzalez for their excellent research assistance and to Marc Painter and the New York Times for generously sharing data on local shelter-in-place orders. Map data copyrighted by OpenStreetMap contributors is openly available on https://www.openstreetmap.org.

## Author Contributions

**Conceptualization:** Christof Brandtner, Luís M. A. Bettencourt, Marc G. Berman.

**Data curation:** Christof Brandtner, Andrew J. Stier.

**Investigation:** Christof Brandtner, Andrew J. Stier.

**Methodology:** Christof Brandtner.

**Supervision:** Luís M. A. Bettencourt.

**Validation:** Luís M. A. Bettencourt, Marc G. Berman.

**Visualization:** Christof Brandtner.

**Writing – original draft:** Christof Brandtner.

**Writing – review & editing:** Christof Brandtner, Luís M. A. Bettencourt, Marc G. Berman, Andrew J. Stier.

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
