## [Decision Letter · Decision Letter 0]

1 Oct 2020

PONE-D-20-23979

Creatures of the state? Metropolitan counties compensated for state inaction in initial U.S. response to COVID-19 pandemic

PLOS ONE

Dear Dr. Brandtner,

Thank you for submitting your manuscript to PLOS ONE. After careful consideration, we feel that it has merit but does not fully meet PLOS ONE’s publication criteria as it currently stands. Therefore, we invite you to submit a revised version of the manuscript that addresses the points raised during the review process.

We look forward to receiving your revised manuscript.

Kind regards,

Haroldo V. Ribeiro

Academic Editor

PLOS ONE

Journal Requirements:

Reviewers' comments:

Reviewer's Responses to Questions

**Comments to the Author**

1. Is the manuscript technically sound, and do the data support the conclusions?

Reviewer #1: Yes

Reviewer #2: Partly

2. Has the statistical analysis been performed appropriately and rigorously? 

Reviewer #1: Yes

Reviewer #2: I Don't Know

3. Have the authors made all data underlying the findings in their manuscript fully available?

Reviewer #1: Yes

Reviewer #2: Yes

4. Is the manuscript presented in an intelligible fashion and written in standard English?

Reviewer #1: Yes

Reviewer #2: Yes

5. Review Comments to the Author

Reviewer #1: The paper “Creatures of the state? Metropolitan counties compensated for state inaction in initial U.S. response to COVID-19 pandemic” investigates the factors that make the local government adopt policies before higher-level governments mandate such measures. The authors developed their analysis using US data considering the adoption of non-pharmaceutical interventions (e.g. shelter-in-place and social distance) to prevent the spread of COVID-19. In order to do so, the authors used a multitude of datasets with socioeconomic and demographic information of counties and the time of adoption of intervention policies by counties and states. They used regression models to find the variables that help or hinder the local implementation of preventive measures before a higher-level of governments take action. This is a very timely topic and it is an interplay between the science of cities and the spreading of innovation study. While I enjoyed very much reading the paper, I had the following concerns:

Major concerns:

1) What is the relationship between the time of adoption of shelter-in-place policies with the time of the first case reported? I would guess there is a correlation between the time since the first case and the adoption of preventive measures.

2) Do counties that were infected earlier also adopted shelter-in-place earlier despite the inaction of the state’s government response?

3) Can the time of first case be incorporated into the regression analysis as a predictor for earlier adoption of intervention measures in addition to the “Cum. cases of COVID-19” and “COVID growth rate”? The number of cases and growth rate depends on the population density and time since the initial case. Perhaps, a variable that accounts for the time between the first case and the implementation of an intervention measure would somehow account for this problem.

4) Additionally, the model could also control for the number of tests since this would directly reflect on the number of cases at an early stage.

Minor observations:

5) Some of the references mentioned in the text were not included in the reference list.

6) It would be helpful to have the equations of the model fully written in the text.

Apart from the concern described above, I strongly recommend the paper for publication in PLOS ONE.

Reviewer #2: The paper "Creatures of the state? Metropolitan counties compensated for state inaction in initial U.S. response to COVID-19 pandemic", presents an analysis of the relative temporal adoptions of shelter-in-place orders by the U.S. counties. When analyzing a series of statistical predictors the authors find that demography is a major factor that determines the timing when shelter-in-place orders are adopted -- Large counties usually started early. The authors also find that political tensions between state and counties could lead to an earlier adoption by the counties when compared to the state.

1) I think the description in Table 1 lacks some information, what is N? Average, S.D. ... of which variables?

2) In the PDF file the images have a very bad resolution, although looking at the original files they are better.

However, Fig. 1 is difficult to understand, even using the high resolution file. I would recommend, perhaps, removing state labels.

3) Although the general message of the paper is clear, I found the text a bit confusing when trying to connect the conclusions with the methods and results. From Figure 2 and Table 2 it is clear that the population plays an important role in determining the time for implementing shelter-in-place orders. However, it is not so clear how the analyzes have changed in Figure 4. Why in Figure 4 is demography no longer important?

4) In Figure 5, how is each line calculated? Is it a nonlinear kernel regression of the data? It is possible to show the data?

Looking at the dashed vertical line, it makes me believe that the day of the first order is the same for all counties.

Was not it supposed to be several dashed lines?

5) I think it would be important to have a histogram of number of days since the shelter-in-place order. This would give

the readers a better view about the scale of the analysis.

6) The conclusion about the role of political tensions is using counties that have implemented shelter-in-place orders

before the state. How many are they? Is it possible to have a list of all of them?

7) The authors used two COVID19 variables: Cumulative cases and the growth rate. However, I suppose that the day of the first case in the county is also important. Does the authors have any idea if the day of first case could be as important as the political tension?

6. PLOS authors have the option to publish the peer review history of their article (what does this mean?). If published, this will include your full peer review and any attached files.

Reviewer #1: No

Reviewer #2: No

---

## [Author Response · Author response to Decision Letter 0]

22 Dec 2020

>> Thank you for the opportunity to revise our manuscript. We were lucky to receive such generous suggestions from the two reviewers and were able to implement them all. Several of the comments, particularly the idea to include an additional covariate for the timing of the first COVID-19 case, gave us an opportunity to offer supplementary analyses. Below and in the attached response letter, we address the reviewers' comments point by point.

Reviewer #1

The paper “Creatures of the state? Metropolitan counties compensated for state inaction in initial U.S. response to COVID-19 pandemic” investigates the factors that make the local government adopt policies before higher-level governments mandate such measures. The authors developed their analysis using US data considering the adoption of non-pharmaceutical interventions (e.g. shelter-in-place and social distance) to prevent the spread of COVID-19. In order to do so, the authors used a multitude of datasets with socioeconomic and demographic information of counties and the time of adoption of intervention policies by counties and states. They used regression models to find the variables that help or hinder the local implementation of preventive measures before a higher-level of governments take action. This is a very timely topic and it is an interplay between the science of cities and the spreading of innovation study. While I enjoyed very much reading the paper, I had the following concerns:

>> Thank you for your excellent suggestions and for your encouraging review.

Major concerns:

1) What is the relationship between the time of adoption of shelter-in-place policies with the time of the first case reported? I would guess there is a correlation between the time since the first case and the adoption of preventive measures.

>> Thank you for raising this important point. We added a variable for the first recorded case in each county as a covariate. Because 716 out of 3,001 counties did not record any cases of COVID-19 prior to April 5, these counties were dropped from the analysis upon inclusion of the timing variable. This is why the overall number of observations differs from our initial submission. We find that our results are robust to the inclusion of this variable (and dropping the 716 counties). We have now made our main analysis, presented in the results section, and have included the results for the full sample of counties (without the date variable) as supporting information.

2) Do counties that were infected earlier also adopted shelter-in-place earlier despite the inaction of the state’s government response?

>> Our analysis confirms your hypothesis that counties that saw earlier infections also experienced shelter-in-place earlier (and generally at a greater likelihood) than counties that saw later infections. However, this effect is not statistically significant for the specific adoption of county orders only (compared to all orders, including state orders; see table 2).

>> Furthermore, figure 4 shows that the time of the first case is significant for Republican counties. This suggests that the functional reason to adopt shelter-in-place orders - i.e. earlier date of first case - were relevant among Republican counties, whereas they had less of an importance among Democratic counties. We discuss the implication of this insight on p. 21.

3) Can the time of first case be incorporated into the regression analysis as a predictor for earlier adoption of intervention measures in addition to the “Cum. cases of COVID-19” and “COVID growth rate”? The number of cases and growth rate depends on the population density and time since the initial case. Perhaps, a variable that accounts for the time between the first case and the implementation of an intervention measure would somehow account for this problem.

>> This suggestion is, again, correct and addressed in our response to points 1 and 2. Thank you once more for pointing out that interventions depend on the timing of first infection.

4) Additionally, the model could also control for the number of tests since this would directly reflect on the number of cases at an early stage.

>> Thank you for this suggestion. To our knowledge, consistent historical data on county-level COVID-19 test capacity are unavailable. However, we were able to include state-level data on the number of tests administered.

>> To address your concern about early-stage cases, our data on COVID-19 cases already contains the earliest cases known in the US. Testing capacity correlates with case numbers later on, but not in the early days of the pandemic (see figure).

Minor observations:

5) Some of the references mentioned in the text were not included in the reference list.

>> Thank you for pointing out this omission of some references. This is now rectified.

6) It would be helpful to have the equations of the model fully written in the text.

>> We concur and have added the equation of the full model to the main text on p. 14–15.

Apart from the concern described above, I strongly recommend the paper for publication in PLOS ONE.

Reviewer #2

The paper "Creatures of the state? Metropolitan counties compensated for state inaction in initial U.S. response to COVID-19 pandemic", presents an analysis of the relative temporal adoptions of shelter-in-place orders by the U.S. counties. When analyzing a series of statistical predictors the authors find that demography is a major factor that determines the timing when shelter-in-place orders are adopted -- Large counties usually started early. The authors also find that political tensions between state and counties could lead to an earlier adoption by the counties when compared to the state.

>> Thank you for your careful reading and generous comments on our paper.

1) I think the description in Table 1 lacks some information, what is N? Average, S.D. ... of which variables?

>> Thanks for raising this issue, which may have been due to a formatting error in the initial submission. Table 1 is a landscape table describing all variables used in the study as well as their source, N (number of observations), mean, standard deviation, minimum, and maximum. Please let us know if the problem persists.

2) In the PDF file the images have a very bad resolution, although looking at the original files they are better. However, Fig. 1 is difficult to understand, even using the high resolution file. I would recommend, perhaps, removing state labels.

>> As you suggest, we removed state labels from the maps in figures 1 and 3 to improve the ease of reading.

>> We regret the low quality of the figures in the PDF. All figures are plotted in print quality and available as direct downloads of high-resolution .tiff files on the PDF page.

3) Although the general message of the paper is clear, I found the text a bit confusing when trying to connect the conclusions with the methods and results. From Figure 2 and Table 2 it is clear that the population plays an important role in determining the time for implementing shelter-in-place orders. However, it is not so clear how the analyzes have changed in Figure 4. Why in Figure 4 is demography no longer important?

>> We appreciate the great question. In our original submission, we used two complementary but different models for these two questions: an event-history analysis looking at the timing of the adoption of an order, and a logistic regression to test whether a county will eventually adopt an order. The result you observed means that a city’s demography matters for timing, but less so for whether an order is ultimately adopted. The research question is most aptly addressed by an event history model, which is why we changed figure 4 to also use a Cox model predicting the timing of adoption.

4) In Figure 5, how is each line calculated? Is it a nonlinear kernel regression of the data? It is possible to show the data? Looking at the dashed vertical line, it makes me believe that the day of the first order is the same for all counties. Was not it supposed to be several dashed lines?

>> Figure 5 shows average levels of distancing and case growth by which policy (county-level, state-level, or none) was first adopted in a given county. You are correct that not all counties adopted orders on the same day (and some did not at all). We therefore added two vertical lines representing the average date of adoption of county-level and state-level policies. We also added histograms of the frequency of new shelter-in-place orders by policy time in order to address the issue that not all counties issued an order this early.

5) I think it would be important to have a histogram of number of days since the shelter-in-place order. This would give the readers a better view about the scale of the analysis.

>> Thank you for this excellent idea. We added these histograms of the date of adoption directly into figure 5.

6) The conclusion about the role of political tensions is using counties that have implemented shelter-in-place orders before the state. How many are they? Is it possible to have a list of all of them?

>> Yes. We included a list of all 163 counties that passed shelter-in-place orders prior to their states in the supporting information. We now also publish all of the data used for the analyses.

>> In the supporting information, we describe in detail what measures we took to ensure the completeness of our data. We are confident that we are able to publish an authoritative list of local shelter-in-place orders from March/April 2020.

7) The authors used two COVID19 variables: Cumulative cases and the growth rate. However, I suppose that the day of the first case in the county is also important. Does the authors have any idea if the day of first case could be as important as the political tension?

>> This good point is in line with Reviewer 1’s suggestion to include the date of the first case as a covariate. We did so in all models and found that a) counties with earlier cases indeed passed earlier orders and that b) our results are entirely robust to the inclusion of this variable.

---

## [Decision Letter · Decision Letter 1]

19 Jan 2021

Creatures of the state? Metropolitan counties compensated for state inaction in initial U.S. response to COVID-19 pandemic

PONE-D-20-23979R1

Dear Dr. Brandtner,

We’re pleased to inform you that your manuscript has been judged scientifically suitable for publication and will be formally accepted for publication once it meets all outstanding technical requirements.

Kind regards,

Haroldo V. Ribeiro

Academic Editor

PLOS ONE

Reviewers' comments:

Reviewer's Responses to Questions

**Comments to the Author**

1. If the authors have adequately addressed your comments raised in a previous round of review and you feel that this manuscript is now acceptable for publication, you may indicate that here to bypass the “Comments to the Author” section, enter your conflict of interest statement in the “Confidential to Editor” section, and submit your "Accept" recommendation.

Reviewer #1: All comments have been addressed

Reviewer #2: All comments have been addressed

2. Is the manuscript technically sound, and do the data support the conclusions?

Reviewer #1: (No Response)

Reviewer #2: Yes

3. Has the statistical analysis been performed appropriately and rigorously? 

Reviewer #1: (No Response)

Reviewer #2: I Don't Know

4. Have the authors made all data underlying the findings in their manuscript fully available?

Reviewer #1: Yes

Reviewer #2: Yes

5. Is the manuscript presented in an intelligible fashion and written in standard English?

Reviewer #1: Yes

Reviewer #2: Yes

6. Review Comments to the Author

Reviewer #1: The authors have fully addressed all the concerns from my previous report. Therefore, I am happy to recommend the paper for publication in PLOS ONE in its current form.

Reviewer #2: I believe the authors have made suitable revisions to their manuscript and I now recommend going ahead with publication.

7. PLOS authors have the option to publish the peer review history of their article (what does this mean?). If published, this will include your full peer review and any attached files.

Reviewer #1: No

Reviewer #2: No

---

## [Editor Report · Acceptance letter]

2 Feb 2021

PONE-D-20-23979R1 

Creatures of the state? Metropolitan counties compensated for state inaction in initial U.S. response to COVID-19 pandemic 

Dear Dr. Brandtner:

I'm pleased to inform you that your manuscript has been deemed suitable for publication in PLOS ONE. Congratulations! Your manuscript is now with our production department. 

Kind regards, 

on behalf of

Dr. Haroldo V. Ribeiro 

Academic Editor

PLOS ONE